# Improving Land Use/Cover Classification Accuracy from Random Forest Feature Importance Selection Based on Synergistic Use of Sentinel Data and Digital Elevation Model in Agriculturally Dominated Landscape

**Sa'ad Ibrahim** 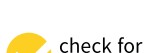

Department of Geography, Adamu Augie College of Education, Argungu PMB 1012, Kebbi State, Nigeria; saad.ibrahim@aacoeargungu.edu.ng

**Abstract:** Land use and land cover (LULC) mapping can be of great help in changing land use decisions, but accurate mapping of LULC categories is challenging, especially in semi-arid areas with extensive farming systems and seasonal vegetation phenology. Machine learning algorithms are now widely used for LULC mapping because they provide analytical capabilities for LULC classification. However, the use of machine learning algorithms to improve classification performance is still being explored. The objective of this study is to investigate how to improve the performance of LULC models to reduce prediction errors. To address this question, the study applied a Random Forest (RF) based feature selection approach using Sentinel-1, -2, and Shuttle Radar Topographic Mission (SRTM) data. Results from RF show that the Sentinel-2 data only achieved an out-of-bag overall accuracy of 84.2%, while the Sentinel-1 and SRTM data achieved 83% and 76.44%, respectively. Classification accuracy improved to 89.1% when Sentinel-2, Sentinel-1 backscatter, and SRTM data were combined. This represents a 4.9% improvement in overall accuracy compared to Sentinel-2 alone and a 6.1% and 12.66% improvement compared to Sentinel-1 and SRTM data, respectively. Further independent validation, based on equally sized stratified random samples, consistently found a 5.3% difference between the Sentinel-2 and the combined datasets. This study demonstrates the importance of the synergy between optical, radar, and elevation data in improving the accuracy of LULC maps. In principle, the LULC maps produced in this study could help decision-makers in a wide range of spatial planning applications.

**Keywords:** land use; land cover; classification; random forest; Sentinel data; SRTM; random forest; feature selection; accuracy; validation



## 1. Introduction

Earth-observing satellite sensor data can be used for land-cover mapping and monitoring, which is essential for estimating land-cover change. The increase in land use and land cover changes (LULC) in natural ecosystems has adversely affected carbon stocks, climate change, and biodiversity, as well as the global climate over the past few decades [1–4]. It is believed that deforestation due to urbanization and agricultural expansion is one of the most critical threats to the environment in the 21st century [5]. The United Nations (U.N.) sustainable development goal (SDG) 15 has emphasized measures to "*protect, restore, and promote sustainable use of terrestrial ecosystems, sustainably manage forests, combat desertification, and halt and reverse land degradation and biodiversity loss*" [6]. Priority is placed on combating desertification, recovering degraded land and soil, particularly in areas affected by desertification, drought, and floods, and combating land degradation by 2030. Satellite Earth observation data offer one of the most reliable options for monitoring land degradation in the context of the SDGs due to their consistency and repeatability at local and large spatial scales. Information about the land cover of a country is an essential part of the planning and

development process. It is useful for environmental reporting [7], assessing the impact of land use on the natural environment [8], conserving biodiversity and habitats [9], mapping population distributions [10], forecasting crops, studying urban heat islands, managing insurance risks, planning telecommunications, and others [11–13].

Even though traditional methods (e.g., field surveying) yield accurate results, they are expensive and inefficient in monitoring large and inaccessible areas. To overcome these limitations, remote sensing scientists have developed analytical tools for detecting, characterizing, parameterizing, and monitoring land variables based on space observations. Remote sensing has experienced rapid advances over the past 40 years. Based on remote sensing technology, data are usually collected across different regions of the electromagnetic spectrum at wide spatiotemporal scales (e.g., the recent Copernicus program/Sentinel missions and the Landsat program/missions- which has been available for over 40 years). Hence, remote sensing provides an interesting option for policymakers to make informed decisions about our environment and also to improve the methodology of assessing ecosystem vulnerability [14,15].

Over the past decades, the scientific community has fully recognized remote sensing/Earth observation data from space for LULC monitoring. These data offer an unparalleled opportunity for large-area measurement and high temporal precision for land cover mapping and monitoring. Today, a large number of global land cover maps are produced (e.g., GLOBCOVER and MODIS land cover products). However, these products have their limitations for regional as well as local assessments due to their low spatial resolution (e.g., 1 km, 250 m), temporal frequency, and inconsistencies in their assigned thematic classes [16]. These limitations primarily occur due to (1) the small number of training data relative to the size of the area being mapped, (2) mismatch definitions/propriety in land cover classification schemes, (3) and the need for a readily and automated algorithm to handle large datasets. In this light, many regional governments have embarked on research projects to provide high and medium-resolution (e.g., 30 m) land cover maps, which are accurate and consistent with their local demands. For example, the operational land cover databases (e.g., the National Land Cover Database for the United States of America (U.S.A.) and the United Kingdom's Land Cover product which is based on the European CORINE land cover mapping scheme [11].

A widespread increase in anthropogenic activities, land use, and land cover changes are occurring at an unprecedented rate, requiring policymakers and stakeholders to pay greater attention to the measures to manage and control environmental degradation. In Nigeria, the threat to environmental sustainability, for example, is encapsulated in the need to ensure the quality of the environment is appropriate for good health and well-being, as well as to protect and utilize the environment and natural resources for the benefit of present and future generations. The policy encourages the compilation of detailed land capability inventories, comprehensive land classifications, assessment of the current land use practices, causes and extent of land degradation, and regulatory framework for sustainable land use [17]. However, despite recent advancements in Earth observation and remote sensing, there is no reliable land LULC for the country. Most of the previous global land cover maps were not also developed based on adequate or training data sets covering Nigeria. And their class labeling and definitions (e.g., International Geosphere-Biosphere Programme) have mixed land cover classes, which are unsuitable for discerning LULC characteristics in Nigeria. Conservation policies in Nigeria have emphasized undertaking land capability classifications based on evolving methods of land evaluation suitable to local conditions.

Land cover monitoring using remotely sensed data involves precise mapping of complex land cover and land use categories, necessitating the employment of strong classification systems [18]. Waske and Braun [19], who compare the ensemble classifiers with approaches such as the maximum likelihood classifier for land cover classification using C-band multi-temporal SAR data, observed that random forest (RF) outperformed maximum likelihood by more than 10%. A comprehensive comparison of machine learning

algorithms has been conducted by Lawrence and Moran (2015) using uniform procedures and 30 distinct datasets. Their results showed that RF had the highest classification accuracy of 73.19% than SVM, which had an accuracy of 62.28%. Of the total 30 classifications, RF was the most accurate in 18 classification scenarios. Talukdar et al. [20] reviewed six machine-learning classifiers for LULC classification using satellite observations. Based on overall accuracy, results indicate that RF is the best machine-learning LULC classifier (0.89, RMSE = 0.006), compared to support vector machine (Kappa = 0.86, RMSE = 0.11), artificial neural network (Kappa = 0.87, RMSE = 0.09), fuzzy adaptive resonance theory-supervised predictive mapping (0.85, RMSE = 0.17), spectral angle mapper (Kappa = 0.84, RMSE = 0.23), and Mahalanobis distance (Kappa = 0.82, RMSE = 0.28). This makes the machine learning algorithm suitable for LULC classification. Furthermore, a recent study by Adugna et al. [21], who compare RF and SVM machine learning methods, found that RF outperformed SVM, yielding overall accuracy (OA) of 0.86 and a kappa (k) statistic of 0.83, respectively, which is 1–2% and 3% higher than the best SVM model.

Nowadays, machine learning technology is used for feature selection to assist in mapping LULC categories. The advantage of RF is its capability for feature selection, which has been proven to improve classification accuracy in previous studies [22–24]. A study by Balzter et al. [11], who developed a method for CORINE Land Cover mapping using RFs, demonstrates the importance of variable selection using Sentinel-1A radar backscatter coefficient at HH and HV polarizations (summer acquisitions) and VV and VH polarization (winter acquisitions) and SRTM Digital Elevation Model Data. The classification out-of-bag error rate was 52.5%, and kappa ($\kappa$) = 0.38 for the Sentinel-1 variables. When the variables generated from the S.R.T.M. data were added, the quality of the classified map was improved substantially, with an out-of-bag error rate of 31.6% (68.4% accuracy) and $\kappa$ = 0.63. R.F. clearly describes the benefits of including variable selection in the land cover classification process in a complex environment [25].

The RF technique is well-established in land remote sensing today. Still, it has not been adequately evaluated by the remote sensing community as compared to more traditional pattern recognition algorithms. In addition, there have been observations about how the importance of variables varies depending on the data and ecosystem in question, necessitating further exploration [23,25,26]. To assist decision-makers in a variety of spatial planning applications (e.g., cropland management, irrigated agriculture intensification, flood vulnerability assessment, water management, or human settlement/resettlement planning in floodplains), the thematic LULC classes were created to represent the local characteristics of the semi-arid region, in Nigeria. Specifically, the objectives of this study were; (1) to evaluate the applicability of an RF classification algorithm for LULC mapping using local class definitions and training data sets in an agriculturally dominated landscape in Nigeria; (2) to assess the contribution of an individual satellite band in the RF model; (3) to improve model performance and reduce prediction errors of LULC maps based on RF feature selection. The novelty of this study is the synergistic use of different sources of satellite data to identify the most important variables to reduce prediction error. Therefore, one of the most important contributions of the work is the methodology developed to improve classification performance. The insights gained in this work to improve model performance and reduce prediction errors not only support policymakers in applying accurate LULC maps in spatial planning but also enrich the methodological system of LULC assessment through machine learning.

## 2. Materials and Methods

### 2.1. The Study Area

This study was conducted in Kebbi state, the northwestern part of Nigeria (Figure 1a,b). This area is located between latitude 4°27′0″–4°54′0″ N of the equator and longitude 4°19′12″–4°48′0″ E of the Greenwich meridian (Figure 1a). This area falls in the Argungu local government and parts of Augie, Birnin-Kebbi, and Gwandu local government areas. The climate in the area is tropical continental, with two distinct seasons,

dry and wet. This is caused by the presence of two contrasting air masses, the tropical continental and the tropical maritime, which originate from the Sahara Desert and the Atlantic Ocean, respectively. The wet season lasts from May to October. The dry season lasts from November to April. The average rainfall is 800 mm. The average temperature is 27 °C which can rise to 40 °C in the summer. Sudan savannah is the predominant vegetation type in the area [27–29]. Geologically, the area is composed of sedimentary rock, primarily undifferentiated sands, gravels, clays (mostly in the upland areas), and floodplains that surround riverine communities [30]. It is, therefore, possible to identify two types of soil in the area: sandy soil for the upland area and clayey and hydromorphic soils for the floodplain area (clay, clay-loam, sandy-loam, loamy sand). The area is mostly characterized by lowland with a few highland areas of up to 344 m, dissected by large flowing rivers (e.g., River Rima) (Figure 1a). The area is characterized by Sudan savannah vegetation type. It includes trees/shrubs (*Pilliostigma reticulatum*, *Combretum nigricans*, *Combretum verticellatum*, *Guira senegalensis Azadirachta indica*, *Piliostima thonningii*, *Guira senegalensis* and grass species (*Borroria scabra*, *Borroria radata*, *Pennisetum peicellatum*, *Pennistum peicellatum*, *Corchorus fascicularis*, *Digitaria horizontalis*, *Lam (karangiya)*, *Commelina forskalei*, *Eragrostis gangetica*, etc.) [28]. These species of plants have different phenological cycles (e.g., leaf flush and senescence period). However, most of these species have their leaf-on up to the end of November.

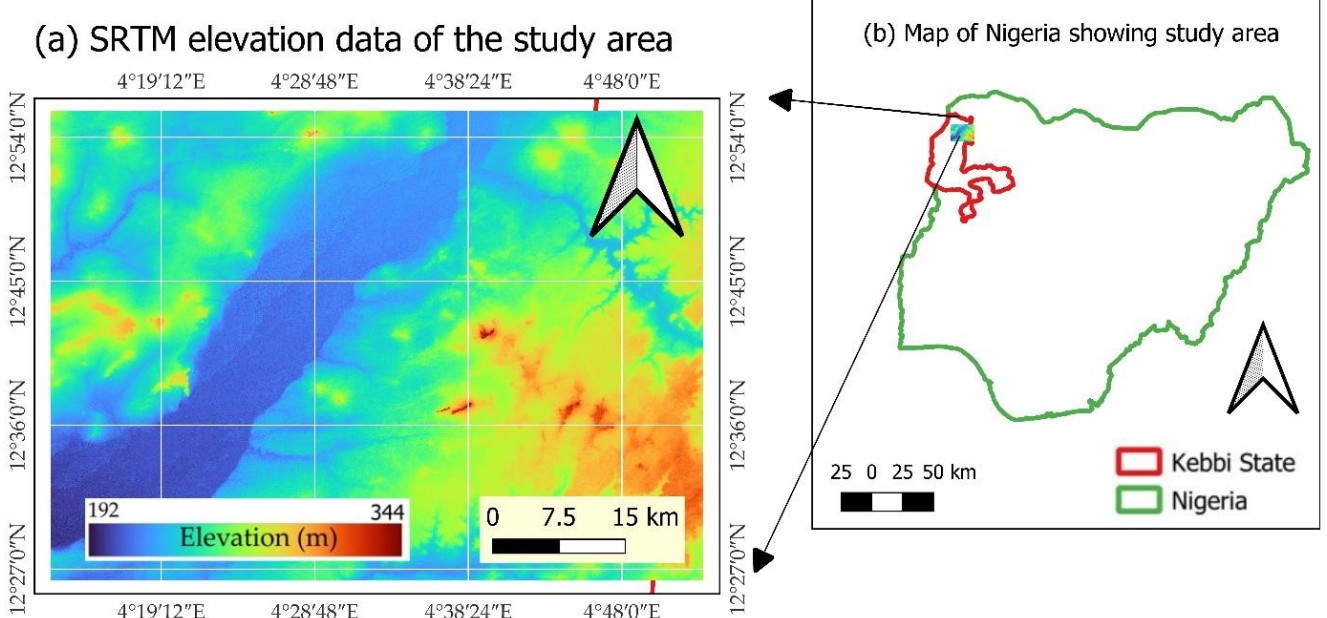

**Figure 1.** Study area (**a**) study area showing the elevation data based on SRTM data, (**b**) Map of Nigeria showing Kebbi State and the location of the study area.

A large number of subsistence farmers use the floodplain area for irrigation activities where rice, wheat, tomatoes, etc., are being cultivated. But the majority of the farmers engaged in rice cultivation. In the upland areas, crops such as millet, guinea corn, legumes, etc., are usually cultivated. The main harvest time for cereals (such as millet) is late October, except for Guinea corn and legumes, which are usually harvested around mid-November. However, most of the cultivated land in the upland region grows millet. The main harvesting season for rice is November and December, depending on the type and timing of planting. Rice grown in the rainy season is fed to some extent by the rain, as it is strongly supported by irrigation activities. Approximately 75% of the people in the area work as farmers and cultivate crops through rain-fed and irrigation practices in the floodplain area [31]. The area is well-known for its contribution to rice production and fishing in Nigeria. The river Rima in Argungu provides an opportunity for tourism-the famous Argungu International Fishing and Cultural Festival on the one hand, and

industrial development- the WACOT rice mill company-employing many thousands of people as well as enhancing rice production in the region. Currently, land use patterns are undergoing various transformations as a result of changing demographic and economic characteristics in the area, creating a wide range of environmental problems. As the land use system continues to undergo rapid changes, there is a need to develop an accurate mapping framework so that an assessment of future land use patterns and the sustainability of land resources may be well-studied.

*2.2. Remote Sensing Data*

2.2.1. Sentinel-1 Normalized Backscatter

Sentinel-1 is a C-band synthetic aperture radar (SAR) satellite mission of the European Copernicus Program. In this study, the Sentinel-1 Analysis Ready Data (ARD) is one of the remote sensing data used as input variables for feature importance selection with the RF Classifier. The Sentinel-1 data were acquired on 4 October 2020, and were downloaded from the Digital Earth Africa website (https://www.digitalearthafrica.org/, accessed on 15 August 2022. Because the wet season occurs from the end of May to the end of October, the image acquisition period was found suitable to capture the phenology of plant species and crops. The data are available in single polarization (VV) and double polarization (VH). In addition, radiometric terrain correction (RTC) was applied to the normalized backscatter [32]. To increase the number of variables in the RF model, two additional variables were created from these polarizations. The mean and total sum of VV and VH were generated and included in the RF model to assess whether these variables could contribute to model performance. In general, data from SAR, such as those from Sentinel-1, provide different and complementary information than that provided by optical remote sensing. A radar signal can penetrate clouds and provide information about the Earth's surface that optical sensors cannot work due to topography, land cover structure, orientation, and moisture characteristics.

2.2.2. Sentinel-2 Surface Reflectance

In addition to other remote sensing data, this study incorporates the Copernicus sentinel-2 multispectral data to map the LULC of the study area. The Sentinel-2 ARD for 17 October 2020, was downloaded from the Digital Earth Africa website at https://maps.digitalearth.africa/, accessed on 15 August 2022. The acquisition period of the imagery was considered useful in capturing the phenology of woody plants, grasses, and crops. The spectral bands used for this study include blue (band 2), green (band 3), red (band 4), red edge (band 5), red edge (band 6), red edge (band 7), NIR (band 8), NIR (band 8a), SWIR1 (band 11) and SWIR2 (band 12). The spatial resolution of these data is 20 m. The data were pre-processed and atmospherically corrected by the providers. Sentinel-2 has promise in LULC mapping in semi-arid/agriculturally dominant landscapes based on RF feature selection [33,34].

2.2.3. SRTM Digital Model Data

It is a collaboration between the National Geospatial-Intelligence Agency and the National Aeronautics and Space Administration (NASA) to provide elevation data at a global scale to produce the most complete high-resolution digital topographic database of Earth using radar data. The 30 m, ArcGRID format was used in this study and is available at http://www2.jpl.nasa.gov/srtm/index.html (accessed on 15 August 2022). Three variables were created from the data. These include elevation, slope, and aspect. This will be used to represent the surface elevation of the study site. The 30 m DEM was upgraded to 20 m through the nearest neighbor interpolation techniques to make it compatible with Sentinel data.

*2.3. Method*

2.3.1. Workflow

This study identified the most critical spectral and topographic variables for enhancing model performance using ARD sentinel data (1 and 2) and SRTM DEM. The RF classification method was used to classify LULC types (river, wetland/flooded, irrigated land, barren, built-up area, tree/shrubland, farmland, and grassland) in the area. The flowchart is shown in Figure 2. The figure illustrates a high-level summary of the processes and procedures employed in this study.

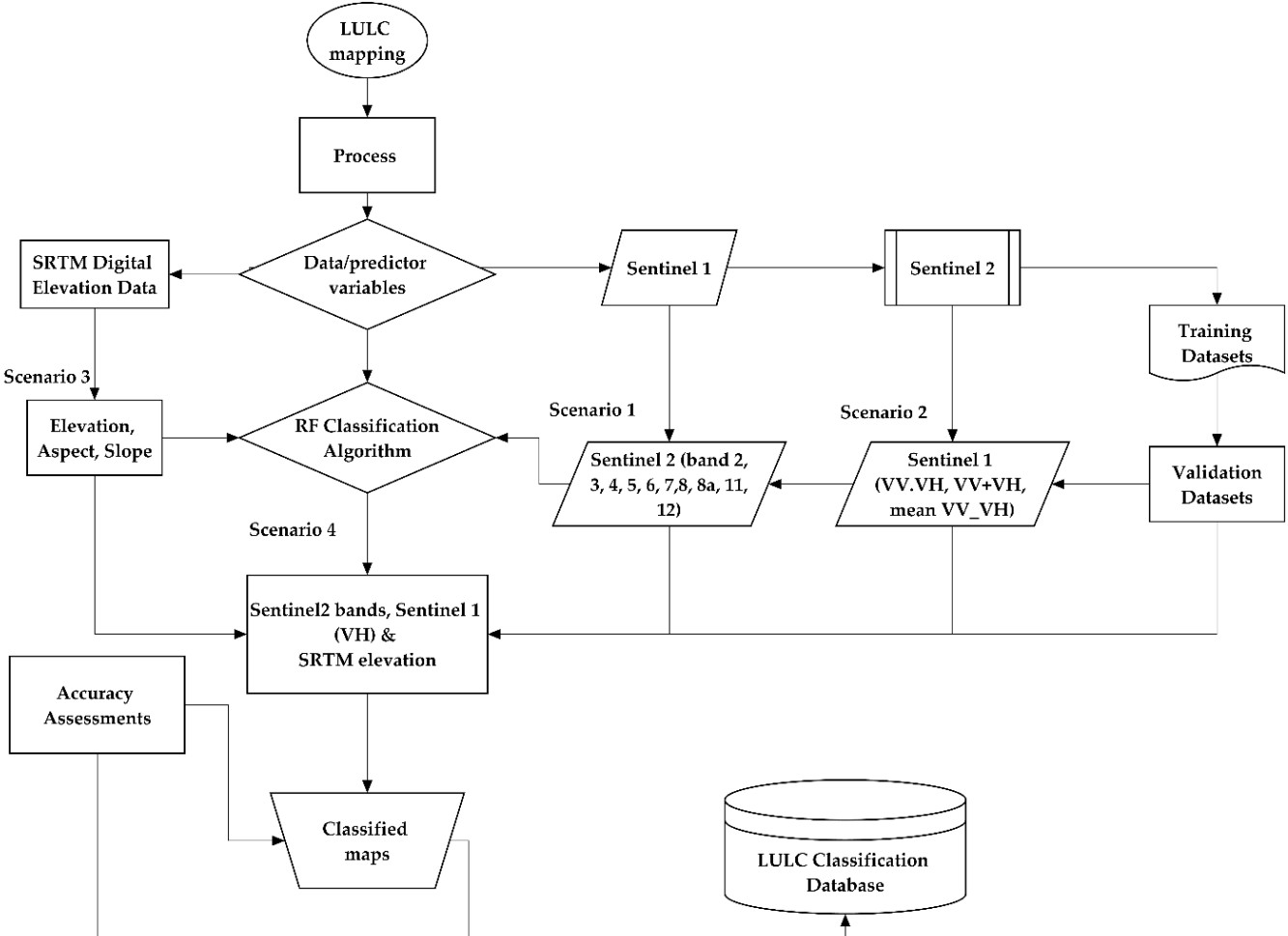

**Figure 2.** The workflow of the processes and approaches implemented in the study.

2.3.2. Resampling

The data used in this study are not on the same spatial resolution. For example, the ARD Sentinel-2 data is 20 m, Sentinel-1 is 25 m, and STRM DEM is 30 m. This makes it necessary to resample these datasets on the same spatial resolution. However, there are different techniques of resampling that are used depending on the problem. The first one is done as a result of a mismatch between the different raster datasets, while the other type is when a raster dataset is converted into a different coordinate system. In this study, the reason for resampling is due to the mismatch between the different data sets mentioned earlier. Three resampling methods, including bi-linear, nearest neighbor, bicubic or cubic convolutional interpolation, are commonly used for resampling raster data [35]. The nearest neighbor resampling approach was found to be more suitable and was therefore used to resample the normalized sentinel backscatter data and data from DEM to 20 m. The nearest neighbor assigns the DN (or another pixel value, like backscatter) based on the input matrix's nearest pixel. It has the advantage of computational simplicity and does

not potentially change input pixel values [35]. Zheng et al. [36], who assessed the effects of different spatial resolution unification schemes and methods on LULC classification, discovered that nearest neighbor interpolation could satisfy the needs of local and regional LULC applications.

### 2.3.3. Feature Importance Selection

Predictor variables were selected based on an understanding of how spectral reflectance varies across surface features and how it contributes to land surface characterization. The electromagnetic spectrum offers a wide range of options for discriminating among various objects. Even within a land cover category, there are variations in the spectral signatures of different electromagnetic spectrum components. For example, in vegetation, light absorption by leaf pigments dominates in the visible wavelength (400–700 nm), whereas leaf pigments are transparent to NIR (700–1300 nm), and leaf absorption is small [37]. Sentinel-2 data, for instance, has 13 bands, each of which contributes differently to the differentiation of the land surface. A unique characteristic of vegetation is its reflectance signature, which is observed by active sensors such as microwaves (e.g., shortwave or longwave radar data). Whether it is day, night, or cloudy, microwave sensors can image any part of the planet. Through this, radar data complement passive optical data in mapping LULC types. Some variables are more relevant for some phenomena than others, depending on the situation at hand. In Figure 3a, normalized backscatter variability is shown for the 8 LULC types being studied. Based on these spectral variations, the LULC types seem to be distinguished across different polarizations (VV, VH, mean VV & VH, VV+VH). Figure 3b shows the spectral curves of the 8 LULC types from Sentinel-2 multi-spectral data. In general, there is a possibility that these classes could be well distinguished by the classifier based on their emittance behavior (Figure 3a,b). The visible wavelengths, especially the blue and green bands, do not discriminate between these LULC types. The LULC classes of red, NIR, and SWIR 1 and 2, however, possess distinct spectral characteristics (Figure 3b).

The complexity of the environment makes it challenging to easily identify which feature is most useful for predicting land cover categories. This is due to the uncertainty as to which of the features will contribute most to the accuracy of classification. Additionally, auxiliary features such as topographic variables are usually included in an RF feature selection to complement spectral data. The ability to combine numerous variables to enable feature selection to better predict outcomes is provided by the RF machine learning feature selection [38,39]. Mean decrease accuracy (MDA) has been recognized as one of the standard procedures for assessing feature importance, which is based on the OOB estimates of the RF model [40,41]. The higher the value, the more important the variable is. To find the most important features for enhancing model performance, sets of scenarios with various features were established, which assessed these features based on individual datasets and in combinations for this investigation (Table 1).

**Table 1.** Predictor variables for the RF feature selection.

| S/No | Scenario 1 | Scenario 2 | Scenario 3 | Scenario 4 |
| | Sentinel-2 Bands | Sentinel-1 Bands | SRTM Data | Combined (Scenario 1, 2 & 3) |
|---|---|---|---|---|
| 1 | Blue | VV | Elevation | Blue |
| 2 | Green | VH | Aspect | Green |
| 3 | Red | Mean (VV & VH) | Slope | Red |
| 4 | NIR_8 | VV+VH | - | NIR_8 |
| 5 | NIR_8a | - | - | NIR_8a |
| 6 | SWIR1 | - | - | SWIR1 |
| 7 | SWIR1 | - | - | SWIR1 |
| 8 | Red edge_1 | - | - | Red edge_1 |
| 9 | Red edge_2 | - | - | - Red edge_2 |
| 10 | Red edge_3 | - | - | - Red edge_3 |
| 11 | - | - | - | VH |
| 12 | - | - | - | Elevation |

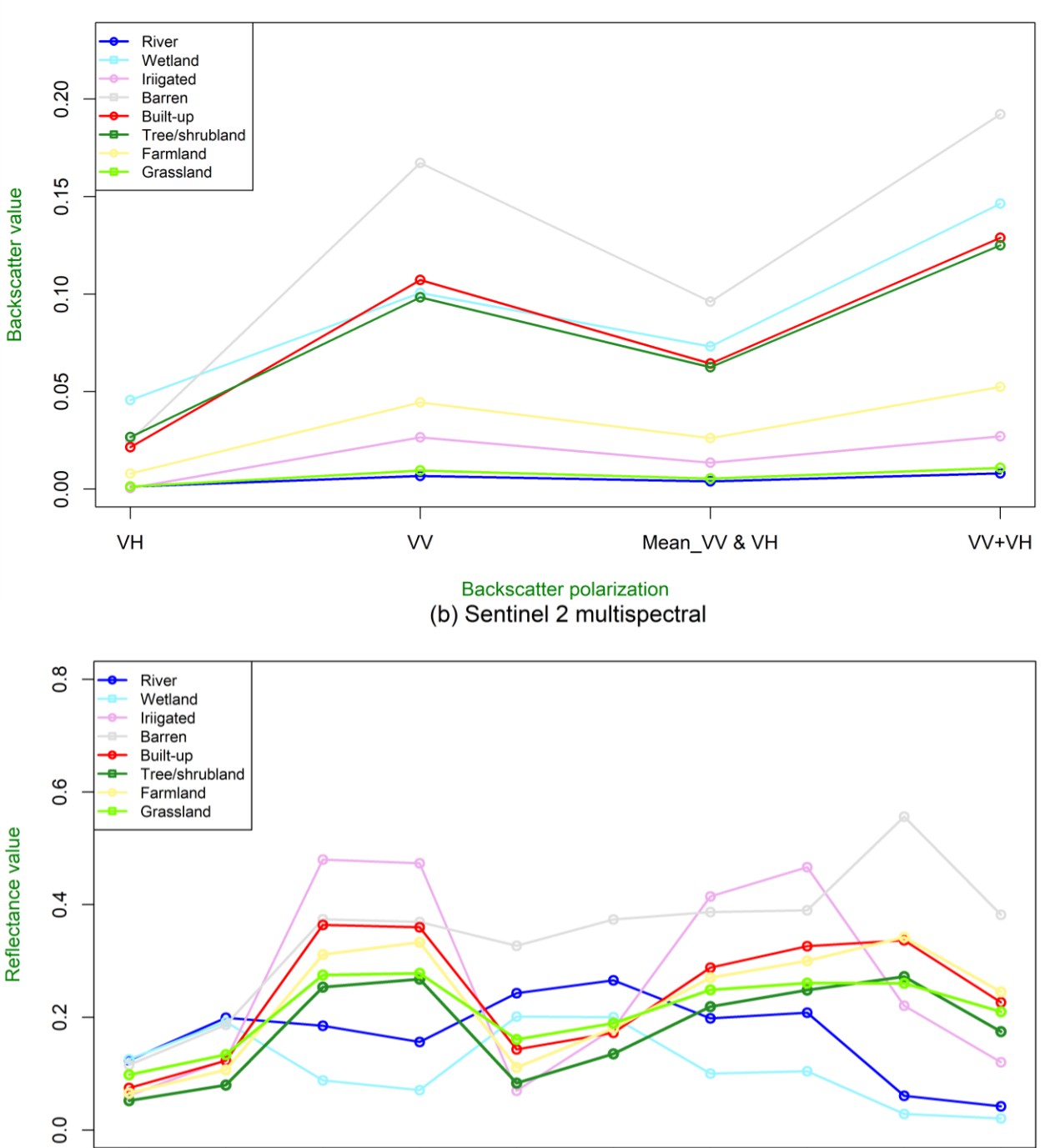

**Figure 3.** (**a**): Spectral curves of the LULC categories derived from the Sentinel-1 normalized backscatter, (**b**) spectral curves of the LULC categories derived from the Sentinel-2 normalized backscatter.

### 2.3.4. Training Data

A previous study evaluating the RF method found that classification accuracy increases with increasing training data [40,41]. This means that accurate classification requires many

training polygons. Therefore, this study digitized 430 polygons for the seven LULC classes using an RGB composite derived from Sentinel-2 and Google Earth.

### 2.3.5. RF Classification

Breiman [24] explains that "RF classification is a combination of tree predictors such that each tree depends on the values of a random vector sampled independently and with the same distribution for all trees in the forest". RF classification of images is based on the principles that construct several decision trees. From the large collection of trees, each tree in the RF splits out a class prediction, and model prediction is performed based on the class with the most votes. This method relies on bootstrap and feature randomness when generating each tree [24]. Liaw and Wiener [42] explain the basic steps in the RF classification procedure as follows:

i.　First, create $n_{tree}$ bootstrap samples using the original data.
ii.　Create an unpruned classification or regression tree for each of the bootstrap samples. At each node, select the best split from a randomly selected subset of the predictors rather than the best among all predictors.
iii.　Assemble the predictions of the ntree trees to predict new data (i.e., majority votes for classification, the average for regression).

In this study, the RF classification was implemented in R statistical software by applying the 'RandomForest' package [42] and other packages such as 'raster' [43], 'sp' [44], 'rgdal' [45], 'sf' [46], 'gstat' [47]. As explained earlier, the LULC in the study area was classified into eight classes. For increased classification accuracy, all pixels in the training data were used for each class. Four scenarios were established based on the predictor variables to determine the most important features and accurate results. To evaluate the performance of classifications, various input features were used:

i.　In the first scenario, only the Sentinel-2 bands were considered as predictor variables.
ii.　In the second scenario, Sentinel-1 normalized backscatter (VV, VH, VV+HH, and the mean of VV and VH) were considered.
iii.　In the third scenario, only the DEM variables (elevation, aspect, and slope) were considered.
iv.　All of the variables considered in scenario 1 and most variables in the second and third scenarios were utilized in the fourth scenario.

### 2.3.6. Out-Of-Bag Error Estimates

The accuracy of the classifications was assessed based on the Out-of-bag (OOB) confusion matrix, which is usually computed internally by the model. The training data is divided into 70%, which is used for the classification, while the remaining 30% is used for the OOB estimation. An estimate of the error rate can be obtained, based on the training data, by the following:

1.　At each bootstrap iteration, predict the data not in the bootstrap sample (what Breiman (2001) calls "out-of-bag", or OOB, data) using the tree grown with the bootstrap sample.
2.　Aggregate the OOB predictions (On average, each data point would be out-of-bag around 30%, which aggregates these predictions). Calculate the error rate and call it the OOB estimate of the error rate.

The OOB confusion matrix, kappa statistics, overall accuracy, and error rate were presented. In addition, class errors were also presented as they can depict LULC type that is more or less accurate and can therefore disentangle the uncertainty associated with overall accuracy based on the classification performance [48].

### 2.3.7. Independent Validation

Researchers are concerned about the reliability of accuracy assessments. This is even though OOB error calculated by the RF is widely recognized as a standard method of error reporting by the scientific community [49,50], some scholars are still of the view that an independent test is required due to bias nature of the RF accuracy assessment [51,52]. It was proposed that cross-

validation can reduce the remaining bias [51]. A well-known phenomenon is RF's preference for predicting classes where the majority of training observations originate [53]. A stratified random sampling of equal size was used for the selection of validation data in this study. The selection of validation shapefile was carried out in R programming software using the 'sp' [44] and 'raster' [43] packages. One hundred points were extracted for each class from the classified maps. However, further confirmation and verification of the individual points were done in QGIS with the help of RGB composite and Google Earth so that the correct class could be assigned to each point data. Similarly, the accuracy assessments of the classified maps were performed in *R* programming software using the validation datasets created earlier. The same R packages were used for accuracy assessments. This cross-validation aims to: (1) complement OOB error estimates of the RF, (2) find out whether two validation results can maintain a consistent pattern, and (3) find out whether sampling the same number of observations in each class could serve as an alternative means of reducing bias.

## 3. Results

### 3.1. Feature Importance Selection

Based on the proposed scenarios for evaluating the feature importance, all variables were put into the RF model, and the importance of each variable was calculated by the score of the accuracy of their contribution to the RF classification (4a/d). The RF classification algorithm is robust as it outputs the contribution of different variables in the model. The feature analyses were carried out for each dataset (Sentinel-1, 2, and topographic variables) separately, and the most important features were selected for the last scenario. Based on the random nature of the model, different scores of importance were derived. The Sentinel-2 variables show the lowest out-of-bag error. Therefore, one of the most important features in the second scenario (VH normalized backscatter) and the third scenario (SRTM elevation) were selected to complement the Sentinel-2 data.

Figure 4a,d shows the mean decrease in accuracy of the model for the four scenarios established and implemented in this study. The greater the accuracy, the more influential the variable is for the classification. Figure 4a shows the mean decrease in accuracy of the first scenario, which uses only the Sentinel-2 data. The mean decrease in accuracy between these variables and for this specific scenario. This means that the difference between the most and least important features is substantial. The blue band contributes the most, followed by the SWIR1 band and the NIR band 8a, NIR band 8a, and SWIR2. In this particular scenario, the red edge bands are the least important features (Figure 4a), with mean decrease accuracy ranging from 40–75. In this scenario, the OOB error estimates are less, meaning that all features have yielded the overall accuracy of the model. These results point to the importance of spectral reflectance property variation and the role of the interacting medium.

The feature importance for the second scenario is shown in Figure 4b. Only the Sentinel-1 normalized backscatter was considered in this scenario. In this scenario, VH normalized backscatter appears as the most important feature compared to VV, Mean, and sum of VV and VH. And there is a wide gap between them. And the VH backscatter has the highest contribution to the model. This does not, however, means that the Sentinel-1 data outperformed the Sentinel-2 data when reference is made to the MDA scores. Although the mean decrease accuracy shows the most important feature based on MDA, the scores do not, however, determine the overall accuracy of the model, especially if two different scenarios are being considered. Feature importance in an RF model depends to a large extent on the combinations of the variables in the model. In the third scenario, only topographic variables were assessed. In the third scenario, elevation has the highest scores, followed by slope and the aspect, aspect. The gap between the elevation scores and that of other topographic features is substantial. This suggests that elevation has an important contribution in discriminating land cover/use categories. In the fourth scenario (Figure 4d), 12 features, selected from across the 1st, 2nd, and 3rd scenarios, were combined to optimize the features and therefore ensure an increase in model performance. In this scenario,

elevation is the most important feature and therefore has the greatest contribution to the classification, followed by the blue band > VH > NIR_8a > SWIR1 > NIR_8 > SWIR2 > green > red and then the red edge band as the least contributors in that order (Figure 4d).

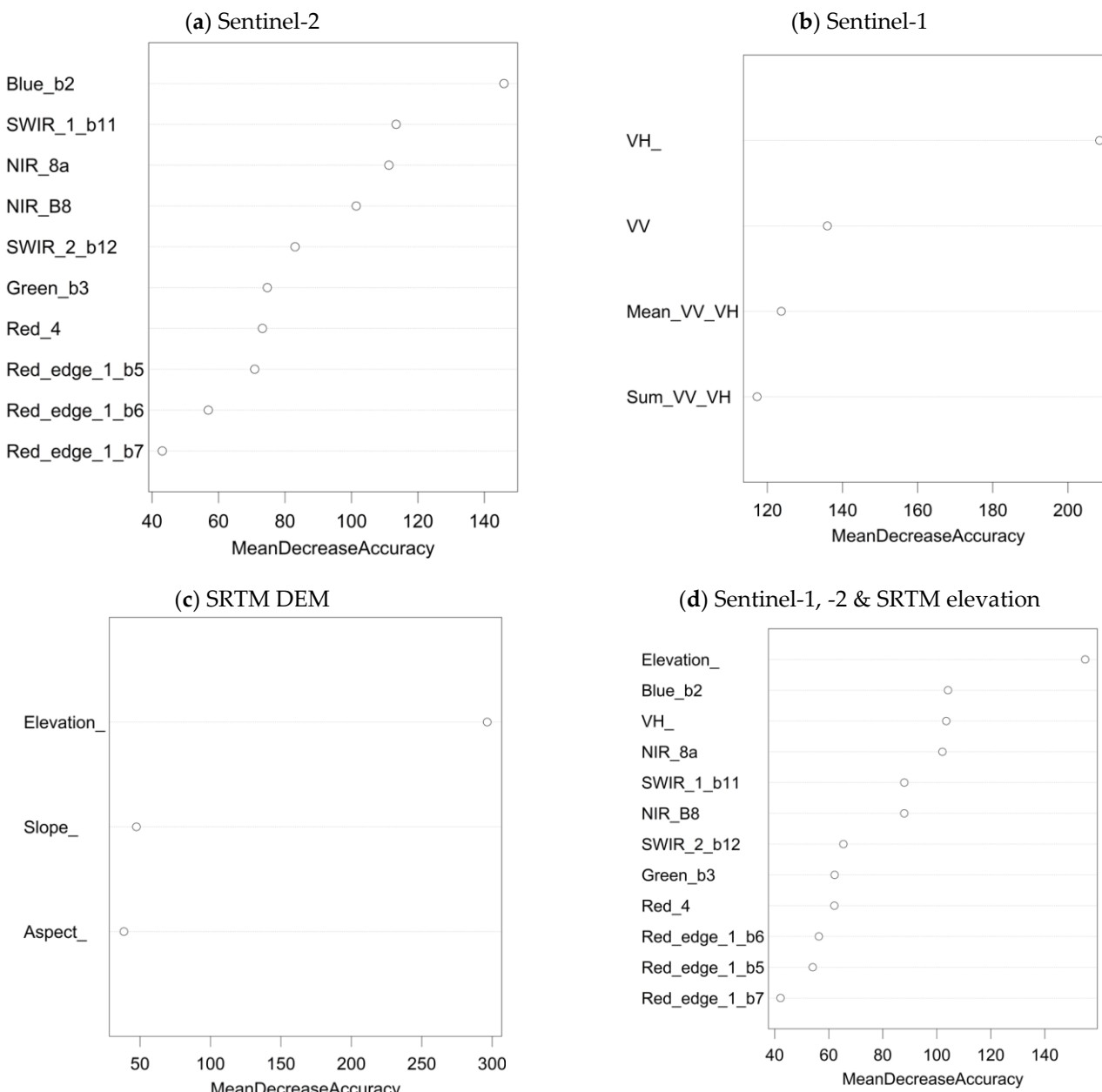

**Figure 4.** The important contribution of the RF feature importance selection-based MDA, (**a**) Scenario 1 (Sentinel-2 only), (**b**) Scenario 2 (Sentinel-1 only), (**c**) Scenario 3 (SRTM digital elevation model data only), (**d**) Scenario 4 (Sentinel-2 bands, Sentinel-1 (VH backscatter) and SRTM elevation data).

*3.2. Out-Of-Bag Error Estimates*

Here, the study presented only the two most accurate classifications (based on Sentinel-2 data and based on a combination of Sentinel-1 VH, Sentinel-2 bands, and SRTM elevation data) based on the most important features of the four classification scenarios explained above. As earlier stated, the purpose of different scenarios implemented in this study was to find out the most important feature for model optimization.

### 3.2.1. Sentinel-2 (Scenario 1) Classification Results

The overall OOB error estimates show that Sentinel-2 bands have an overall accuracy of 84.2%, an OOB error rate of 15.8%, and k = 0.4 (Table 2). Going by the overall accuracy, one can infer that the classification results for these data are highly accurate. However, as expected, and as it is most common to many classification results, there are omission as well as commission errors in the classification results. The RF model provides an error rate for each class of the land cover/use category. Irrigated land has the least class error (5.3%), while grassland has the highest (21%), followed by tree/shrubland (18.30%) and then farmlands (17.49%) (Table 2). This means that there is a probability that pixels classified in these categories may not be the actual land cover on the ground. For example, the spectral signatures of farmlands resembled that of grassland and farmland. This led to confusion and misclassifications of these land categories. The misclassification is confirmed, given that these classes recorded the highest errors. The river and the wetland were overestimated, given the spatial resolution of the datasets. Barren and built-up areas confuse each other with barren land in the floodplain region classified as built-up owing to their spectral similarity.

**Table 2.** OOB confusion matrix for Sentinel-2 (Scenario 1) classification results. Overall error rate = 15.8%, Overall accuracy = 84.2%, κ = 0.38.

| LULC Category | 1 | 2 | 3 | 4 | 5 | 6 | 7 | 8 | Row Total | Class Error (%) |
|---|---|---|---|---|---|---|---|---|---|---|
| River | 1145 | 71 | 46 | 11 | 20 | 0 | 0 | 0 | 1293 | 11.45 |
| wetland | 84 | 776 | 1 | 6 | 11 | 0 | 1 | 0 | 879 | 11.72 |
| irrigated land | 9 | 0 | 443 | 1 | 0 | 14 | 0 | 1 | 468 | 5.34 |
| Built-up | 1 | 3 | 7 | 1263 | 29 | 0 | 20 | 0 | 1323 | 4.54 |
| Barren | 5 | 2 | 0 | 135 | 7516 | 374 | 187 | 9 | 8228 | 8.65 |
| Tree/shrubland | 0 | 0 | 108 | 2 | 56 | 1978 | 208 | 69 | 2421 | 18.30 |
| Farmland | 0 | 2 | 57 | 425 | 581 | 3702 | 32,745 | 2173 | 39,685 | 17.49 |
| Grassland | 2 | 0 | 3 | 0 | 0 | 15 | 21 | 147 | 188 | 21.81 |
| Column total | 1246 | 854 | 665 | 1843 | 8213 | 6083 | 33,182 | 2399 | 54,485 | |

### 3.2.2. Sentinel-1 (Scenario 2) Classification Results

The overall OOB error estimates show that Sentinel 1 backscatter has an overall accuracy of 83%, an OOB error rate of 17%, and k = 0.22 (Table 3). Going by the overall accuracy, the result is encouraging. However, the RF's class error shows otherwise. Only farmland achieved a class error of less than 5%, while other classes recorded not less than 43%. The Sent1nel backscatter does not separate the different classes proposed in this study.

**Table 3.** OOB confusion matrix for Sentinel 1 (Scenario 2) classification results. Overall error rate = 17%, Overall accuracy = 83%, κ = 0.22.

| LULC Category | 1 | 2 | 3 | 4 | 5 | 6 | 7 | 8 | Row Total | Class Error (%) |
|---|---|---|---|---|---|---|---|---|---|---|
| River | 285 | 118 | 2 | 33 | 48 | 12 | 794 | 1 | 1293 | 77.95 |
| wetland | 80 | 496 | 0 | 56 | 7 | 0 | 239 | 0 | 878 | 43.50 |
| irrigated land | 0 | 1 | 109 | 2 | 67 | 5 | 283 | 1 | 468 | 76.70 |
| Built-up | 37 | 54 | 0 | 505 | 31 | 2 | 692 | 0 | 1321 | 61.77 |
| Barren | 11 | 5 | 14 | 16 | 5419 | 97 | 2663 | 3 | 8228 | 34.13 |
| Tree/shrubland | 8 | 0 | 5 | 3 | 222 | 703 | 1479 | 1 | 2421 | 70.96 |
| Farmland | 133 | 63 | 37 | 277 | 1034 | 299 | 37,826 | 15 | 39,685 | 4.68 |
| Grassland | 1 | 0 | 0 | 0 | 28 | 0 | 134 | 25 | 188 | 86.70 |
| Column total | 555 | 737 | 167 | 892 | 6856 | 1118 | 44,110 | 46 | 54,482 | |

### 3.2.3. SRTM Elevation Data (Scenario 3) Classification Results

The overall OOB error estimates show that SRTM data (elevation, aspect, and slope) have an overall accuracy of 76.44% and an OOB error rate of 23. 56%, and k = 0.10 (Table 4). Going by the overall accuracy, the result is encouraging. On the contrary, the RF's class error shows otherwise. Only farmland achieved a class error of less than 4%, while other classes recorded not less than 60%. The SRTM data do not separate the different LULC

classes proposed in this study. However, it always shows a good result when it is combined with other multi-spectral and radar data.

**Table 4.** OOB confusion matrix for SRTM data (Scenario 2) classification results. Overall error rate = 23.56%, Overall accuracy = 76.44%, κ = 0.10.

| LULC Category | 1 | 2 | 3 | 4 | 5 | 6 | 7 | 8 | Row Total | Class Error (%) |
|---|---|---|---|---|---|---|---|---|---|---|
| River | 476 | 152 | 34 | 12 | 404 | 1 | 114 | 1 | 1194 | 60.13 |
| wetland | 259 | 169 | 52 | 8 | 202 | 0 | 121 | 0 | 811 | 79.16 |
| irrigated land | 75 | 51 | 80 | 6 | 130 | 1 | 96 | 0 | 439 | 81.78 |
| Built-up | 62 | 28 | 19 | 125 | 184 | 14 | 830 | 1 | 1263 | 90.10 |
| Barren | 92 | 73 | 35 | 24 | 1881 | 14 | 5363 | 0 | 7482 | 74.86 |
| Tree/shrubland | 1 | 5 | 2 | 8 | 109 | 67 | 2039 | 0 | 2231 | 97.00 |
| Farmland | 39 | 52 | 50 | 46 | 960 | 45 | 36,109 | 0 | 37,301 | 3.20 |
| Grassland | 3 | 0 | 0 | 3 | 11 | 1 | 160 | 0 | 178 | 99.00 |
| Column total | 1007 | 530 | 272 | 232 | 3881 | 143 | 44,832 | 2 | 50,899 | |

### 3.2.4. Sentinel-1, 2, VH Backscatter and SRTM Elevation Data (Scenario 4) Classification Results

Table 5 presents the confusion matrix for the classification results in scenario 4. Regarding Sentinel-2 classification results, a consistent pattern has been maintained by the combinations of Sentinel-1, -2, VH backscatter, and SRTM elevation data but with improvement in the classification accuracy (Table 5). The overall accuracy is 89.8% and a κ value of 0.4 (Table 5). This show an increase of 4.9% and 5.3% compared to Sentinel-2 for overall accuracy and kappa statistics, respectively. Similarly, grassland has the highest class error (18.09%), which is still 3% lower than that of Sentinel 2. Grassland was followed by wetland/flooded area (12.19%), farmland (11.8), and tree/shrubland (12.02%). The class error for tree/shrubland is the lowest for this scenario and is 6.28% lower than that obtained in scenario 2 (Table 5). Generally, the addition of the other two features (VH normalized backscatter and elevation data) has improved the overall accuracy of the classification.

**Table 5.** Out-of-bag confusion matrix of the Sentinel-1, -2, VH backscatter, and SRTM elevation data (Scenario 4) classification. Overall OOB error rate = 10.9%, Overall accuracy = 89.1%, κ = 0.4.

| LULC Category | 1 | 2 | 3 | 4 | 5 | 6 | 7 | 8 | Row Total | Class Error (%) |
|---|---|---|---|---|---|---|---|---|---|---|
| River | 1178 | 50 | 37 | 11 | 17 | 0 | 0 | 0 | 1293 | 8.89 |
| wetland | 88 | 771 | 0 | 1 | 17 | 0 | 1 | 0 | 878 | 12.19 |
| irrigated land | 7 | 0 | 457 | 4 | 0 | 0 | 0 | 0 | 468 | 2.35 |
| Built-up | 1 | 3 | 8 | 1276 | 14 | 0 | 19 | 0 | 1321 | 3.41 |
| Barren | 0 | 0 | 1 | 64 | 7617 | 297 | 242 | 7 | 8228 | 7.43 |
| Tree/shrubland | 0 | 0 | 60 | 2 | 41 | 2130 | 156 | 32 | 2421 | 12.02 |
| Farmland | 0 | 0 | 18 | 451 | 682 | 2143 | 34,965 | 1426 | 39,685 | 11.89 |
| Grassland | 2 | 0 | 2 | 0 | 0 | 12 | 18 | 154 | 188 | 18.09 |
| Column total | 1276 | 824 | 583 | 1809 | 8388 | 4582 | 35,401 | 1619 | 54,482 | |

### 3.3. Independent Validation

To complement the validation results obtained in an RF model (which uses 30% of the training datasets), another independent validation was carried out to compare the two scenarios (Tables 6 and 7).

**Table 6.** Confusion matrix for Sentinel-2 accuracy assessment (Scenario 2) classification. Overall error rate = 30.1%, Overall accuracy = 69.9%, κ = 0.66.

| LULC Category | 1 | 2 | 3 | 4 | 5 | 6 | 7 | 8 | Row Total | Accuracy (%) | |
|---|---|---|---|---|---|---|---|---|---|---|---|
| | | | | | | | | | | Producer's | User's |
| River | 91 | 20 | 16 | 6 | 0 | 0 | 0 | 0 | 133 | 90.1 | 68.4 |
| wetland | 10 | 79 | 1 | 26 | 0 | 1 | 0 | 0 | 117 | 79 | 67.5 |
| irrigated land | 0 | 0 | 72 | 0 | 0 | 16 | 0 | 0 | 88 | 72 | 81.8 |
| Built-up | 0 | 0 | 3 | 46 | 2 | 0 | 1 | 6 | 58 | 46.9 | 79.3 |
| Barren | 0 | 0 | 0 | 5 | 90 | 7 | 4 | 6 | 112 | 90.9 | 80.4 |
| Tree/shrubland | 0 | 0 | 6 | 0 | 5 | 66 | 20 | 22 | 119 | 64.7 | 55.5 |
| Farmland | 0 | 1 | 0 | 15 | 2 | 5 | 75 | 25 | 123 | 75 | 61 |
| Grassland | 0 | 0 | 2 | 0 | 0 | 7 | 0 | 39 | 48 | 39.8 | 81.2 |
| Column total | 101 | 100 | 100 | 98 | 99 | 102 | 100 | 98 | 798 | | |

**Table 7.** Confusion matrix for Sentinel 2, 1 (VH backscatter) and SRTM elevation data (Scenario 4) classification. Overall error rate = 24.8%, Overall accuracy = 75.2%, κ = 0.71.

| LULC Category | 1 | 2 | 3 | 4 | 5 | 6 | 7 | 8 | Row Total | Accuracy (%) | |
|---|---|---|---|---|---|---|---|---|---|---|---|
| | | | | | | | | | | Producer's | User's |
| River | 93 | 13 | 13 | 6 | 0 | 0 | 0 | 0 | 125 | 92.1 | 74.4 |
| Wetland | 8 | 86 | 2 | 26 | 0 | 5 | 0 | 0 | 127 | 86 | 67.7 |
| Irrigated land | 0 | 0 | 75 | 0 | 0 | 7 | 0 | 0 | 82 | 75 | 91.5 |
| Built-up | 0 | 0 | 3 | 48 | 2 | 0 | 2 | 7 | 62 | 49 | 77.4 |
| Barren | 0 | 0 | 0 | 3 | 91 | 5 | 4 | 5 | 108 | 91.9 | 84.3 |
| Tree/shrubland | 0 | 0 | 3 | 13 | 5 | 78 | 13 | 12 | 124 | 76.5 | 62.9 |
| Farmland | 0 | 1 | 1 | 2 | 1 | 3 | 79 | 24 | 111 | 79 | 71.2 |
| Grassland | 0 | 0 | 3 | 0 | 0 | 4 | 2 | 50 | 59 | 51 | 84.7 |
| Column total | 101 | 100 | 100 | 98 | 99 | 102 | 100 | 98 | 798 | | |

### 3.3.1. Sentinel-2 Accuracy Assessments (Scenario 1)

The confusion matrix for the Sentinel-2 data classification (Scenario 2) is presented in Table 6. In this validation, an equal-size stratified random sampling was used for the selection of validation datasets (800 points, 100 points each for the eight land cover/use categories) were used. The study reports an overall accuracy of 69.9%, an error rate of error, and a κ value of 0.66. Except for grassland and built-up area, all LULC categories achieved producer and user accuracy of more than 55%.

### 3.3.2. Sentinel-1, -2, VH Backscatter and SRTM Elevation Data (Scenario 4) Accuracy Assessment

Table 7 indicates the accuracy assessment of the RF classification results conducted based on an independent validation for Sentinel-1 and -2, VH backscatter, and SRTM elevation data. In this scenario, the study observed an overall accuracy of 75.2% and a k-value of 0.71. The study noticed an improvement in terms of model performance compared to when Sentinel-2 only was used. In this scenario, all classes recorded the user's accuracy of at least 62%.

### 3.4. LULC Maps and Area Covered by Each LULC Category

Table 8 shows the area proportion as extracted from the LULC maps obtained from the RF classifications for Sentinel-2 only (Figure 5a) and the combination of Sentinel-1, -2, VH backscatter, and the SRTM elevation (Figure 5b), which are presented in Figure 5a,e, the two most accurate LULC maps. Quantitatively, it is obvious that cultivated areas dominate the landscape, with farmland occupying close to 3000 km$^2$ of the land. On the other hand, wetlands, rivers, and grassland constitute a smaller proportion. The maps show the types of LULC categories that exist in the area. Visually, the maps show that cropland (upland agriculture) predominates in the area. Despite the predominance of upland agricultural land use, the RF model's ability to discern across classes makes the maps even more intriguing. LULC categories like river, wetland/flooded, irrigated land, and grassland

are relatively modest in comparison to other LULC categories, but the amount of specific information that comes from the classification is detailed and relatively accurate. The floodplain area was clearly distinguished from the LULC in upland areas. This has been achieved in both scenarios. Figure 5c,e shows a full-resolution comparison between the two maps based on Sentinel-2 RGB color composite. In comparison to RGB, there is a clear difference between how the two scenarios classified the LULC classes. Sentinel-2 only seems to have observed more barren land than the combined datasets (Figure 5c,f). On the other hand, the map produced from the combined datasets shows a more vegetated area. These differences occur as a result of variations in the spectral reflectance signature of the land categories. But the use of sentinel backscatter and elevation data has helped to adjust the confusion between classes, which led to improved classification performance.

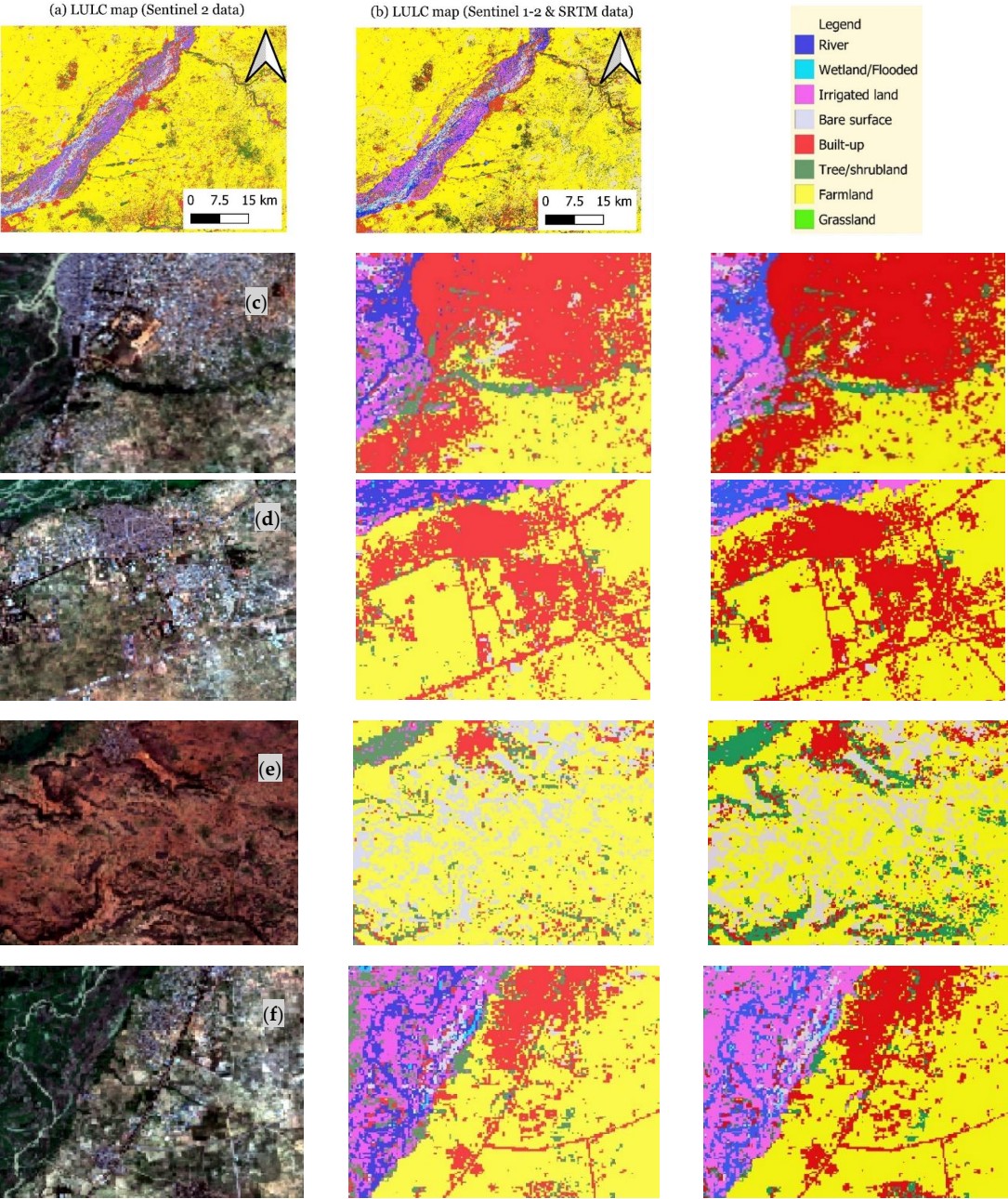

**Figure 5.** LULC maps (**a**) LULC map derived from Scenario 1 (Sentinel-2) (**b**) Scenario 4 (LULC map derived from Sentinel-1, -2 and SRTM elevation data), (**c**–**f**) four areas extracted from the RGB composite of Sentinel-2 and their corresponding locations in the two maps presented in (**a**) and (**b**).

**Table 8.** Area proportion (km$^2$) for each LULC category.

| LULC Category | Sentinel-2 | Sentinel-1, -2 & SRTM Elevation |
|---|---|---|
| River | 58.42 | 55.38 |
| wetland | 22.91 | 25.34 |
| irrigated land | 140.07 | 183.54 |
| Built-up | 110.67 | 142.28 |
| Barren | 481.55 | 425.56 |
| Tree/shrubland | 305.80 | 310.62 |
| Farmland | 2990.68 | 2964.54 |
| Grassland | 3.63 | 2.27 |

## 4. Discussion

This study reports on the production of LULC maps based on random feature selection to evaluate its application in an agriculturally dominated landscape in Nigeria. The potential of using Sentinel-1 optical data, Copernicus Sentinel-1 c radar backscatter, and SRTM topographic variables were investigated to ascertain whether this synergy could improve classification accuracy. The general findings that emerged from this study suggest that: (1) the application of RF classification appears promising in this ecosystem; (2) the use of multiple remote sensing and environmental variables is an important contribution to quantitative remote sensing applications; (3) feature selection methods can improve classification accuracy; however, the evaluation of classification accuracy requires a thorough and critical assessments.

The mapping performed in this study was guided by the RF feature selection procedure based on the ranking of MDA as a function of OOB error estimates. The contribution of each satellite band varies. Some bands make a better contribution than others. What makes these results interesting is the procedure used to test each data set individually and then in combinations. Interestingly, the most important bands also provide the largest spectral differences between classes, except for the normalized backscatter polarizations, where the spread between classes is not very large, but this is similar behavior observed for the Sentinel-2 blue band (Figure 3a,b). Among the Sentinel-2 data variables, the blue, SWIR1, and NIR bands were found to be the most important variables (Figure 4a). Similar behavior for the SWIR1 and blue spectral bands of Sentinel-2 has been observed in previous vegetation, tree species, and crop mapping studies [54,55]. ED Chaves et al. [56] have explained that Sentinel's two SWIR bands are very sensitive to chlorophyll content, which allows them to distinguish different vegetation types and determine classification accuracy for LULC. ED Chaves, CA Picoli, and D. Sanches [56] stated that Sentinel's two SWIR bands are very sensitive to chlorophyll content, allowing them to distinguish different vegetation types and determine classification accuracy for LULC. In addition, visible and shortwave infrared wavelengths are known for their spectral variations, which can explain variations caused by chlorophyll content, soil type, and soil color [57].

Using the normalized backscatter, the VH polarization has the highest rank, which is due to the combinations of the different polarizations (Figure 4b). For the topographic SRTM variables, the elevation data had the highest rank (Figure 4c). The stand-alone classification results for the Sentinel-1 data (Table 3), as well as for the topographic SRTM variables (Table 4), achieved very low accuracy compared to the Sentinel-2 data (Table 2). Therefore, the synergy between VH, elevation data, and Sentinel spectral bands was evaluated to see if the accuracy of the model could be improved. The ranking of the most important variables shows that elevation, blue band, VH, NIR8a, and SWIR1 are the five most important variables (Figure 4d). Elevation makes the largest contribution to the classification. These results are consistent with a recent study by Zhao, Zhu, Wei, Fang, Zhang, Yan, Liu, Zhao, and Wu [57], the only difference being that they do not include radar backscatter as one of their input variables. This study highlights the importance of altitude and radar backscatter data with Sentinel-2 data to improve the classification accuracy of LULC.

The accuracy of the classified maps in this study suggests that it is reasonable to use different remote sensing data for LULC, as has been done in many previous studies. Based on the OOB error estimates, two scenarios were considered the most important, so the comparison is limited to these. The overall OOB classification results for Sentinel-2 data show an overall accuracy of 84.2% and a κ of 0.38, with the lowest and highest class errors for classification at 4.54% and 21.81% for built-up areas and grassland, respectively (Table 2). This level of accuracy is achieved by the Sentinel-2 data alone, further emphasizing their applicability in LULC mapping in this particular ecosystem. However, when the SRTM elevation data and VH backscatter were added to the Sentinel-2 spectral bands, the overall accuracy was 89.1%, and the κ value was 0.4, an increase of 4.9% in overall accuracy (Table 3) compared to the Sentinel-2 data alone. The lowest classification error was found in the irrigated areas, with only 2.15%, while the highest error occurred in the grassland areas (18.09%), which in this case were reduced by 3.72%. The cultivated areas had a class error of 3.41%, which is a further reduction of 1.13% compared to the Sentinel-2 data. For trees and shrubland, the study found a 6.28% difference between the sentinel data and their combination with elevation and VH backscatter data.

This is not independent of the role of the elevation and backscatter data in the overall performance of the model. The topography of the area is heterogeneous, and some of the classes are located in the floodplain, which is typically undulating compared to developed and agricultural areas. Several studies have shown the importance of elevation data to increase the accuracy of the classified map [11,26,40,58]. In the same vein, radar backscatter was found to improve model performance because it can normalize or reduce the effects of the atmosphere, topography, instrument noise, etc., to provide consistent spatial and temporal comparisons [59]. The results from this study are consistent with Meneghini [60], who evaluates the synergy between the Sentinel-1 and Sentinel-2 data for land cover classification. Their results show an overall accuracy of 74% and 78% for Sentinel-2 (Only) and in combination with Sentinel-1 data, respectively. Similarly, several studies have reported the importance of synergy between sentinel-1 and -2 data for increasing model performance for biomass estimation [61], crop type classification [62], irrigation mapping [63], and land cover mapping [64,65].

It has been observed that in a setting in which there is a strong interest in predicting observations from the smaller classes, sampling the same number of observations from each class for validation is a promising alternative [53]. Moreover, one of the objectives of this study was to compare the validation of OOB error estimates of the RF normally performed internally by the model with another independent validation (external) which was performed based on equal-size random stratified sampling using 100 polygons for each LULC category. The overall accuracy of the classification results were 69.9% and 75.2% for Sentinel's 2 data only and the combination of the same data with VH backscatter and elevation data, respectively. The difference between the two is 5.3% which conformed to the OOB estimates of errors even though the overall accuracy obtained from the OOB is higher. The consistency of these two validation results manifested even within the class error. Similar to OOB estimates of error, grassland had the lowest producer's accuracy with an 11.2% difference between the Sentinel's data only and in combination with VH and elevation data based on the independent validation. In this context, the estimates from the OOB are, therefore, reliable since the two validation results have maintained a consistent pattern. The only difference between the two is in kappa statistics, where the external validation shows higher kappa ($k = 0.71$, Table 7) than the estimates from the RF internal validation ($k = 0.4$, Table 3). This is one of the advantages of a balanced setting for applying the equalized stratified random sampling for validation [66], but balancing may not always be possible due to costs or other reasons [4]. But kappa is not a measure of accuracy but of agreement beyond chance, and chance correction is rarely needed [67,68]. The comparison results obtained in this study are consistent with findings by Adelabu et al. [69], who tested the reliability of the internal accuracy assessments of the RF for classifying tree defoliation levels using different validation methods. One of the most

important deductions that can be made in this context is that where only the RF approach is applied to the LULC classification, independent validation is not necessary because validation requires a large number of points, and therefore manual class labeling based on external validation is tedious and time-consuming. The findings of this current study provide insights into the reliability and applicability of OOB error estimates.

One of the limitations of this study is the lack of reference ground truth datasets from a field campaign. Although this study relied on RGB composite images and Google Earth data for the selection of training and validation datasets, it should be noted that such datasets are well-acknowledged as a source of training and validation for land cover mapping [70,71]. Furthermore, a comparison of the quantitative and qualitative results showed that the LULC categories are detailed and very accurate (Tables 2, 3 and 6–8 and Figures 4 and 5). The area estimated from the two most accurate results shows that there is extensive agricultural land. The two maps show slight differences for the area of different LULC categories. The study, however, acknowledged the confusion between the barren land and the built-up areas, which occurred primarily due to the presence of settlements in or near the floodplain areas, in addition to the similarity of the spectral reflectance signatures of these LULC classes. Moreover, the difference between the spectral reflectance signatures between the barren land on the upland and in the floodplain probably led to the underestimation of barren land in the upland areas. However, the class error for barren is minimal, as observed for the RF internal validation (7.43%) as well as for independent validation (producer's accuracy = 91.9% and user's accuracy = 84.3%). From these results, it is obvious that further research in this particular ecosystem may require the need to incorporate vegetation (e.g., NDVI), bare soil indices (e.g., modified normalized difference bare-land index), and water indices (e.g., Modified Normalized Difference Water Index) to improve classification performance. The study also noted confusion between the river network and wetlands. Earlier reports indicated that significant flooding occurred in the area on October 1 [72,73]. At this time, the volume of rivers usually increases, and flooding is easily possible when the amount of rainfall is significant, and the dams along these rivers have been opened. These floods have left many people homeless and severely damaged agricultural land and crops. Future research could focus on flood vulnerability assessment based on change detection using sentinel data. In this situation, flood vulnerability mapping can provide critical information to assess flood risk in the region. Policymakers could be well informed about the risk and thus develop appropriate mitigation strategies based on the severity of the impacts [74,75].

Similarly, the study observed confusion between the grassland and farmland. Mapping LULC with Sentinel-2 data in the semi-arid region is quite promising [34] but challenging because most crops are planted during the rainy season, and their growing season is in July and August, during which the cloud cover is high in the area. And the reliance on dry season imagery may not be feasible as there is a transition from cropland to barren land in the area, especially from early November. Since cropland makes up most of the LULC in the area, this is not the most appropriate time for LULC mapping. This study minimized this problem by integrating Sentinel-1 and -2 data in early and mid-October. Van Tricht, Gobin, Gilliams, and Piccard [63] demonstrate the importance of choosing phenological cycles for crop mapping based on the synergy between the sentinel-1 and -2 data using an RF classifier for increasing model performance. Similarly, many studies demonstrated the importance of Sentinel-1 and -2 for rice mapping in a lowland area [76], mapping paddy rice [77], and mapping Maize Areas in heterogeneous agriculture [78] based on RF. By understanding this trade-off, the current study can help in the selection of datasets and periods for LULC classification with specific applications to agricultural landscapes in semi-arid regions. Although cloud cover may result in a lack of cloud-free imagery in this region, a potential area for further research would be to examine crop and vegetation phenological cycles and by incorporating more variables from the Sentinel-1 data during the rainy season to minimize the challenge of cloud cover.

## 5. Conclusions

This paper proposed LULC mapping by applying an RF classifier to Sentinel-1, -2, and SRTM digital elevation data to evaluate its applicability based on local class definitions and training datasets in an agricultural landscape in Nigeria. The main objective was to develop a methodology to improve model performance and reduce prediction error in LULC classifications. A feature selection method (RF) was implemented to evaluate the contribution of individual bands based on a standard OOB error estimate (MDA). The study showed that the combination of spectral bands, backscatter, and topographic features could improve classification accuracy. The results show that among the variables in the sentinel-2 data, the blue, SWIR1, and NIR bands are the most important variables. Using the normalized backscatter, the VH polarization has the highest rank, which is due to the combination of the different polarizations. For the SRTM topographic variables, the elevation data had the highest rank. The ranking of the most important variables when combining the different data sets shows that height, blue band, VH backscatter, NIR8a, and SWIR1 are the five most important variables.

The overall OOB classification results for Sentinel-2 data show an overall accuracy of 84.2%, with the lowest and highest class errors for classification of 4.54% and 21.81% for built-up areas and grassland, respectively. This level of accuracy is achieved by the Sentinel-2 data alone (scenario 1), further emphasizing its applicability in LULC mapping in this particular ecosystem. On the other hand, the class errors for Sentinel-1 (scenario 2) and SRTM data (scenario 3) show high-class errors. However, when the Sentinel-1, -2, and SRTM elevation data were added to the model, the overall accuracy was 89.1%. This represents a 4.9% improvement in overall accuracy compared to Sentinel-2 alone and a 6.1% and 12.66% improvement compared to Sentinel-1 and SRTM data, respectively. The lowest classification error was found in the irrigated areas at only 2.15%. In comparison, the highest error occurred in the grassland areas (18.09%), which in this case were reduced by 3.72% compared to the Sentinel data alone. According to the study, there was a 6.28% difference between sentinel data and their combination with elevation and VH backscatter data for trees and shrubland. The results of an independent validation based on an equal-size random sampling of 800 points are consistent with OOB error estimates. The study shows how the synergy of optical, radar, and elevation data can significantly improve LULC map accuracy. Based on these results, LULC maps could be used in a broad range of spatial planning applications.

**Funding:** This research was funded by the Tertiary Education Trust Fund (TETFUND) through the Institution Based Research (IBR).

**Data Availability Statement:** Not applicable.

**Acknowledgments:** TETFUND's (IBR) support is gratefully acknowledged. I would also like to give a special thanks to the R Core team for accessing various library packages for data analyses.

**Conflicts of Interest:** The author declares no conflict of interest.

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
