# Peer review of "Improving Land Use/Cover Classification Accuracy from Random Forest Feature Importance Selection Based on Synergistic Use of Sentinel Data and Digital Elevation Model in Agriculturally Dominated Landscape"

_agriculture, doi:10.3390/agriculture13010098_

Round 1

Reviewer 1 Report

The manuscript titled “Improving Land Use Land Cover Classification Accuracy from Random Forest Feature Importance Selection Based on Synergistic Use of Sentinel data and digital elevation model in Agriculturally Dominated Landscape” explored the method of the synergy between optical, radar, and elevation data in improving the accuracy of LULC maps, which is significant for decision-makers in a wide range of spatial planning applications. However, there were some issues to be improve such as 1) the study focused on the area with agriculturally dominated landscape, you had better pay more attention on the agricultural landscape, not only the Farmland ,Grassland and  Tree/shrub land. Authors should give/consider the time of acquisition of images used in the study and the ability to distinguish crop categories in conjunction with the seasonal characteristics of the study area. 2) In this study, different source data were used,  The 30 m D.E.M. will be upgraded to 30 m through the nearest 213 neighbor interpolation techniques in order to make it compatible with Sentinel data”, in Section2.2.3. SRTM digital model data”. Please  provide detailed strategies for resampling remote sensing data from different sources. 3) There is no ground truth data in this study to verify the classification results. It is suggested to calculate the area or proportion of different land use types in different classification results, and discuss and analyze the applicable scope or region of the classification method in this paper.

Reviewer 2 Report

Please refer to the PDF file.

Reviewer 3 Report

The authors presented a workflow for generating LULC maps from sentinel 1 and 2, as well as SRTM DEM based on the Random Forest classification. As for me, the topic is so interesting, but the novelties are low and limited. However, I recommend some general and meaningful feedbacks for the author to improve his work.  

1- The Random forest is widely used for LULC mapping. The question is how to improve the quality of this algorithm in LULC mapping. The author can find a lot of papers that use the RF as the core of classification which can be found in the following list: 

https://doi.org/10.1080/10106049.2022.2123959

https://doi.org/10.3390/rs14153778

2- To clearly show the novelties, please explain more about the third part of the objective in the last paragraph of the paper. 

3- The author mainly explains the important contribution of the RF feature importance selection-based fitness function (here is MDA). Please change the title of the paper, if possible. 

4- I can't find good explanations about senario1-... in the figure. 

Round 2

Reviewer 1 Report

The revised version of the paper has met the publication requirements and is approved for publication.

Reviewer 2 Report

I am happy that the author has corrected all the minor mistakes.

Reviewer 3 Report

The authors have addressed all my comments for this paper and answered the technical questions I have for this method.